# NO Synthesis but Not Apoptosis, Mitosis or Inflammation Can Explain Correlations between Flow Directionality and Paracellular Permeability of Cultured Endothelium

**DOI:** 10.3390/ijms23158076

**Published:** 2022-07-22

**Authors:** Mean Ghim, Sung-Wook Yang, Kamilah R. Z. David, Joel Eustaquio, Christina M. Warboys, Peter D. Weinberg

**Affiliations:** Department of Bioengineering, Imperial College London, London SW7 2AZ, UK; mean.ghim@yale.edu (M.G.); josh.sw.yang92@gmail.com (S.-W.Y.); kamilah.r.z.david@gmail.com (K.R.Z.D.); j.eustaquio@imperial.ac.uk (J.E.); cwarboys@rvc.ac.uk (C.M.W.)

**Keywords:** leaky junction, cell death, cell division, FITC-avidin, transport, orbital shaker, atherosclerosis, hotspots, tricellular junction

## Abstract

Haemodynamic wall shear stress varies from site to site within the arterial system and is thought to cause local variation in endothelial permeability to macromolecules. Our aim was to investigate mechanisms underlying the changes in paracellular permeability caused by different patterns of shear stress in long-term culture. We used the swirling well system and a substrate-binding tracer that permits visualisation of transport at the cellular level. Permeability increased in the centre of swirled wells, where flow is highly multidirectional, and decreased towards the edge, where flow is more uniaxial, compared to static controls. Overall, there was a reduction in permeability. There were also decreases in early- and late-stage apoptosis, proliferation and mitosis, and there were significant correlations between the first three and permeability when considering variation from the centre to the edge under flow. However, data from static controls did not fit the same relation, and a cell-by-cell analysis showed that <5% of uptake under shear was associated with each of these events. Nuclear translocation of NF-κB p65 increased and then decreased with the duration of applied shear, as did permeability, but the spatial correlation between them was not significant. Application of an NO synthase inhibitor abolished the overall decrease in permeability caused by chronic shear and the difference in permeability between the centre and the edge of the well. Hence, shear and paracellular permeability appear to be linked by NO synthesis and not by apoptosis, mitosis or inflammation. The effect was mediated by an increase in transport through tricellular junctions.

## 1. Introduction

Haemodynamic wall shear stress (WSS) and endothelial permeability both vary from site to site within the arterial system. The variation in WSS is likely to account, at least in part, for the variation in permeability [1], and this influence has been associated with the initiation of atherosclerosis: the disease is hypothesised to develop in regions where WSS is low in magnitude [2], oscillatory [3] or multidirectional [4] and where macromolecules cross the endothelium particularly rapidly [5,6,7]. However, there has been long-standing debate on the shear-sensitive route by which macromolecules pass through the endothelium into the arterial wall.

One widely investigated possibility is that macromolecules cross the endothelial barrier predominantly via particularly leaky junctions between neighbouring cells [8]. Leaky junctions can arise when an endothelial cell is dying or dividing, and an association in vivo between foci of elevated permeability and dying or dividing cells has been demonstrated by direct visualisation of tracer uptake [9,10,11,12,13,14]. The foci tend to occur more frequently at atherosclerosis-prone sites, but the relationship with flow is hard to assess in such studies. Surgical interventions and in vitro experiments have demonstrated that endothelial cell apoptosis and proliferation are increased by the types of flow implicated in atherogenesis [15,16,17] and that they are associated with elevated permeability [18,19,20,21,22]. However, none of these studies directly visualised transport around apoptotic and proliferating cells; the association was investigated indirectly, by assessing the correlation between permeability and the number of such cells.

We previously used direct visualisation of tracer uptake to examine how putatively pro- and anti-atherosclerotic flows influence macromolecule permeability in vitro [23]. Our data suggest that low-density lipoprotein (LDL)-sized tracers cross the endothelium via a vesicular mechanism; the rate of transport is determined by the extent of multidirectional flow, an effect that is mediated by endothelial secretion of follistatin-like 1 [24]. High-density lipoprotein (HDL)- and albumin-sized tracers cross by a paracellular pathway [23]. The junctions where three or more cells meet (here termed tricellular junctions for simplicity) play a major role, and they are also affected by multidirectional flow [23,25]; however, the mechanisms have not been established.

Here we investigated mechanisms linking the extent of multidirectional flow to paracellular permeability. We initially focused on the roles of early- and late-stage apoptosis, proliferation and mitosis. Since the data failed to demonstrate an unequivocal link between any of these phenomena and elevated permeability at the cellular scale, we additionally studied the roles of proinflammatory changes and decreased NO synthesis, both of which are early events in atherosclerosis, influenced by WSS and known to affect transport [26,27,28,29,30,31]. The data did not support a role for inflammation, but an inhibitor of NO synthesis abolished effects of multidirectional flow on permeability.

## 2. Results

### 2.1. Effect of Shear on Permeability

Chronic application of shear reduced FITC-avidin tracer accumulation underneath the monolayer by 30% overall (Figure 1A). Figure 1B shows FITC-avidin accumulation as a function of radial distance from the centre of the well. There was no effect of distance for static monolayers. With shear, tracer accumulation was elevated in the central region of the well, where putatively proatherogenic flow occurs, and reduced towards the edge of the well, where putatively antiatherogenic flow occurs. Figure 1C shows representative tile scans of FITC-avidin accumulation along the radius of static and sheared wells. A gradient is visible in the sheared case but not in the static one.

### 2.2. Effect of Shear on Apoptosis, Proliferation and Mitosis

Chronic application of shear reduced the number of cells in early- and late-stage apoptosis by 95% and 97% overall (Figure 2A and Figure 2B, respectively). The number of apoptotic cells varied widely between donor aortas under static conditions but became more consistent under shear. There was no obvious effect of radial distance on early- or late-stage apoptosis under static conditions (Figure 2C,D). When shear was applied, regional differences appeared: the number of early-stage apoptotic cells between radial distances of 5 and 8 mm was 57% lower than between 0 and 3 mm (Figure 2C,E; *p* = 0.03). The equivalent figure for late-stage apoptotic cells, although not significant, was 41% (Figure 2D,F; *p* = *ns*).

Chronic shear reduced the number of proliferating cells by 77% (Figure 2G) and the number of mitotic cells by 55% (Figure 2H). The number of proliferating cells was consistent with a previous study [32] and, as in that study, independent of radial distance, both under static and shear conditions (Figure 2I). The number of mitotic cells (Figure 2J) was also consistent with a previous study [20] where mitotic figures were identified by haematoxylin staining. There was, again, no clear trend with radial distance, with or without shear, but the absolute number of events was small, and, hence, the data are noisy.

### 2.3. Correlation of Permeability with the Number of Apoptotic, Proliferating and Mitotic Cells

Tracer accumulation at different radial distances from the centre of the well under shear is plotted against the frequency of early- and late-stage apoptosis, proliferation and mitosis at the same radial location in Figure 3A–D, respectively. The correlation was positive and significant for the first three cases, but not for mitosis (for which the data were noisy—see above). 

Note that values obtained under static conditions, which would not follow the same relation, are excluded from Figure 3.

### 2.4. Colocalisation Studies

Confocal images of static monolayers stained for early-stage apoptosis, late-stage apoptosis, proliferation and mitosis are shown in the first column of Figure 4A–D. Subsequent columns show tracer accumulation, endothelial nuclei and an overlay of all three channels for the same field of view. In all cases, the large majority of FITC-avidin hotspots (green) were separated by many nuclei from the cellular events indicated by immunostaining (red, yellow, turquoise or purple).

Quantitatively, hotspots in the vicinity of early-stage apoptotic, late-stage apoptotic and proliferating cells each accounted for less than 15% of total tracer accumulation under static conditions and for less than 5% after chronic exposure to shear (Figure 5A–C). The equivalent figures for mitotic cells were 2% and 1% (Figure 5D). Collectively, they accounted for approximately 32% under static conditions and approximately 7% under shear.

The distribution of the total tracer accumulation with radial distance from the centre of the well did not correlate with the distributions of tracer accumulation due to early-stage apoptotic, late-stage apoptotic, proliferating or mitotic cells (Figure 5E, Table 1).

Few instances of high FITC-avidin accumulation were seen around early-stage or late-stage apoptotic cells. For proliferating cells, high tracer accumulation was seen only when the cell nucleus was densely concentrated, suggesting that the cell was undergoing mitosis. High tracer accumulation was always seen around cells that stained positively for mitosis itself (e.g., Figure 4D).

Quantitatively, <15% of early-stage apoptotic or proliferating cells and <30% of late-stage apoptotic cells were associated with spots of tracer uptake under static conditions (Figure 5F–H). However, approximately 100% of mitotic cells were associated with spots (Figure 5I). In the first three cases, the percentages tended to drop when shear was applied but the differences were not significant. For mitosis, the percentage was unchanged.

### 2.5. Effect of Shear on Inflammation and Tracer Uptake

The nuclear/cytoplasmic ratio of NF-κB p65 was approximately uniform with radial distance from the centre of the well under static conditions and after 7 days of shear, and was approximately the same under both conditions (Figure 6A). There was no significant correlation between translocation and tracer accumulation at different radial locations under shear (data not shown).

Since NF-κB p65 translocation shows cyclical changes in response to inflammatory stimuli, we also determined its behaviour and the corresponding values of tracer accumulation over time. Tracer accumulation, averaged across the well, tended to increase above static levels after 2 h of shear exposure, although the effect was not significant. It returned to the baseline at 4 h and then continued to decrease until reaching approximately two thirds of its static level (Figure 6B), the latter value being consistent with Figure 1. The decrease from the initial values reached statistical significance by day 1. Translocation showed a broadly similar trend (except that the values at 4 h were proportionately lower) but none of the changes reached significance (Figure 6C).

Equivalent data subdivided according to radial distance from the centre of the well are shown in Figure 6D,E. There was no significant correlation between tracer accumulation under sheared monolayers and levels of NF-κB p65 translocation at different radial distances for any duration of shear except 2 h (Table 2).

### 2.6. Effect of L-NAME on Permeability with and without Shear

The NO synthase inhibitor L-NAME had no effect on tracer accumulation averaged across the well under static conditions (Figure 7A). Chronic application of shear reduced tracer accumulation as before. Under chronic shear, prior treatment with L-NAME increased tracer accumulation to a level that was greater not only than in the monolayers without L-NAME but also than in the monolayers under static conditions (Figure 7A). The lack of effect of L-NAME under static conditions suggests an absence of non-specific toxic effects at the dose employed.

When the data were subdivided according to radial distance from the centre of the well, L-NAME had no significant effect on permeability at any location under static conditions (Figure 7B). Under shear, however, L-NAME not only increased permeability at all locations but abolished the influence of the radial location (Figure 7C).

Further analysis of the effect of L-NAME examined the role of tricellular junctions. The fraction of tricellular junctions associated with spots of FITC-avidin uptake was approximately 25% at all radial locations under static conditions, and was unaffected by L-NAME (Figure 8A). When chronic shear was applied, the fraction was approximately unchanged at the centre of the well but decreased towards the edge. Adding L-NAME increased the fraction to around 35–40% at all radial locations (Figure 8B).

Tricellular junctions accounted for approximately 80% of tracer accumulation under all conditions: the proportion was unaffected by radial location, shear or L-NAME (Figure 8C,D). The mean tracer accumulation under each leaky tricellular junctions was unaffected by radial distance or L-NAME under static conditions (Figure 8E), but under shear it was higher at the centre than at the edge of the well. L-NAME abolished the latter difference, giving values for tracer accumulation that were slightly above those in the centre of the well in the absence of L-NAME everywhere (Figure 8F). Example confocal images acquired for the different conditions are given in Figure 8G.

## 3. Discussion

In a previous study [23], we showed that transendothelial transport of FITC-labelled avidin and neutravidin occurs by a paracellular route. We also showed that when wells containing endothelial monolayers were swirled on an orbital shaker, permeability to FITC-neutravidin increased in the centre and decreased towards the edge. A subsequent study [25] showed that most FITC-avidin is transported through tricellular junctions and that the number of leaky tricellular junctions increases—and that each of these has higher permeability—at the centre than at the edge of swirled wells.

Here, we sought to investigate the mechanisms underlying these phenomena, and specifically the roles of apoptosis, proliferation, mitosis, proinflammatory transcription factors and NO production, all of which have been implicated in determining endothelial permeability. The study employed arterial endothelial cells of low passage number [23], swirling well configurations for which shear stresses have been characterised by computational fluid dynamics [33], chronic rather than acute exposure to flow [20,23], staining for both early- and late-stage apoptosis, and an avidin-based tracer that binds to the biotinylated endothelial substrate and whose accumulation can consequently be compared to the state of the individual cells immediately overlying it [23,34].

We found, as before, that permeability to FITC-avidin was increased in the centre of the well and decreased towards the edge of the well by swirling. The rate at which permeability fell with increasing distance from the centre was initially slow but increased after 2 mm; permeability reached a plateau by 5 mm that was maintained out to 8 mm, the maximum distance examined. As previously discussed [23], this pattern of permeability does not strongly resemble the trend in either the time-averaged WSS, which is approximately constant, or the oscillatory shear index (OSI), which decreases in a nearly linear fashion [33]. There is also a poor correlation with transverse WSS (transWSS, a measure of multidirectionality which averages those components of WSS during one cycle that act at right angles to the mean WSS direction [35]): it, like the time-averaged WSS, is approximately constant over the 0–8 mm range.

Nevertheless, the trend in permeability is well-explained by multidirectionality: it closely resembles the trend in the transWSSmin, which is the transverse shear experienced by cells that are oriented so as to minimise it rather than oriented with the mean shear vector. Our recent data [36] show that endothelial cells do not align with the mean shear vector; under the flow conditions used here, they align between the mean WSS direction and the direction that minimises the transWSS. They are therefore expected to experience a pattern of transverse shear that is intermediate between the transWSS (constant) and the transWSSmin (sigmoid with respect to radial distance)—that is, a sigmoid trend, albeit with a less extreme range than the transWSSmin.

After chronic exposure to shear, there were significant correlations of permeability with the rates of early-stage apoptosis, late-stage apoptosis and proliferation when variation with radial distance from the centre of the well was considered. There was also a convincing correspondence between changes in permeability and changes in NF-κB p65 translocation, both averaged across the well, as a function of time after shear stress was first imposed. There was no significant correlation between radial variation in permeability and the rate of mitosis, but absolute numbers of mitotic events were low and the data were consequently noisy.

Despite these correlations, the data, when taken as a whole, are not consistent with any of the cellular events dominantly determining the pattern of permeability across the well. First, early-stage apoptosis, late-stage apoptosis, proliferation and mitosis were all drastically reduced by chronic shear stress when averaged across the well, whereas permeability was only moderately decreased; for this reason, the data from the static controls do not fit with the correlations obtained under shear. Notably, apoptosis in the centre of the well was reduced by an order of magnitude by shear whereas permeability was significantly increased by it. Second, NF-κB p65 translocation was little changed from the static control levels by chronic shear and, when examining radial variation, there was no correlation between translocation and permeability at any duration of imposed shear beyond 2 h. Third, and most importantly, when individual patches of tracer accumulation in swirled wells were compared with the cells above them, the fraction of the total accumulation that was associated with early-stage apoptosis, late-stage apoptosis, proliferation or mitosis did not exceed 5% for any of these events.

Thus, when comparing spatial variation in permeability with spatial variation in apoptosis and proliferation, there was a significant correlation at the millimetre scale but not at the cellular level. Similarly, permeability and nuclear NF-κB staining showed comparable temporal trends when their levels were averaged over the whole well but no correlation at the cellular level. These apparent contradictions, not previously reported, can be explained if flow characteristics determine permeability and cellular events independently of one another.

It is well-established that endothelial NO production influences permeability and that these effects depend on WSS, albeit in complex ways. The present study found that inhibiting NO production was without effect under static conditions, perhaps because the cells produced insufficient NO to alter permeability, but, consistent with our earlier study [20], that it prevented a decrease in average permeability caused by chronic shear. A new finding was that L-NAME also abolished the difference in permeability between the centre and the edge of the well under chronic shear. Note that NO production may also be responsible for the variation in cellular events: it is well-established that endothelial NO production is antiapoptotic, antiproliferative and anti-inflammatory.

Combining measurements of permeability at a cellular scale with automated semantic segmentation of tricellular junctions enabled identification of novel mechanisms by which shear and NO affect permeability. Shear reduced the number of patent tricellular junctions in non-central locations and slightly reduced the permeability of each one, together accounting for the reduced permeability seen in these regions. At the centre of the well, shear did not noticeably increase the number of patent tricellular junctions but did increase the permeability of each one. L-NAME had no effect on the number of patent tricellular junctions or their permeability under static conditions, but under shear it led to high values of both variables and eliminated differences from the centre to the edge.

Although the results support the view that it is NO production which mediates the effects of shear stress characteristics on paracellular permeability, there are unexplained aspects of the data. We had expected that L-NAME would simply abolish the effects of chronic shear on permeability, bringing permeability back to the constant level seen at all radial locations under static conditions. L-NAME did produce a constant level, but it was almost double the one seen under static conditions. Hence, permeability in the absence of NO production seems higher in swirled wells than in static wells, suggesting that an additional factor may be involved.

Next, we compared our results with previous in vitro studies concerning the roles of dying and dividing cells. Cancel et al. [18] showed that 44% of albumin transport under static conditions occurred through the leaky junctions associated with such cells. This figure was obtained from data collected in the presence of a pressure-driven transendothelial water flux and by using a mathematical model of a three-pore system. Nevertheless, the value is comparable with the 32% we obtained by direct visualisation of the similarly sized FITC-avidin, albeit without the pressure gradient. The figure we obtained under the more physiological condition of chronic shear, however, was only around 7%. Note that there is evidence for receptor-mediated transcellular transport of albumin [37] and that this is unlikely to occur for FITC-avidin.

For LDL transport, Cancel et al. [18] obtained a value of at least 90%, and further evidence supporting a substantial role for mitosis and apoptosis in LDL transport was obtained by manipulating these phenomena [19,21,22]. If such results were confirmed by direct visualisation, the greater importance of apoptosis and mitosis would presumably be explained by the much larger size of the LDL particle and its lower diffusion coefficient. However, a different, transcellular route was observed in our study of an LDL-sized quantum dot tracer [23].

The role of endothelial mitosis and apoptosis in creating “hotspots” of elevated uptake has also been studied in vivo. For example, Lin et al. showed that 99% of mitotic cells [9] and 63% of dead or dying cells [11] were associated with hotspots of albumin uptake in the rat aorta. The number for mitosis is identical to the figure we obtained for FITC-avidin uptake under shear; the number for dead and dying cells is substantially higher than the one we obtained for apoptosis, perhaps reflecting the wider definition: Lin et al. identified dead and dying cells by their uptake of IgG. What is missing from such studies is an estimate of the fraction of the total uptake caused by apoptotic and mitotic cells. We recently addressed that issue using en face confocal microscopy to detect uptake of rhodamine-labelled albumin in the vicinity of aortic branches and found that uptake in hotspots was an order of magnitude lower than the total uptake [38]. That is consistent with the present study where proliferating, mitotic and apoptotic cells together accounted for less than 10% of the total uptake under shear. Again, the importance of hotspots could be greater for larger molecules such as LDL.

Finally, we consider the physiological and pathological relevance of our in vitro data. Inhibiting NO production has complex effects on transport in the intact artery—like permeability, flow multidirectionality and the prevalence of atheromata [39], the effects differ between regions upstream and downstream of branch points, and change with age [30,40,41]. Reducing NO may not always increase paracellular permeability. Furthermore, it is hard to assert that raised paracellular permeability is necessarily proatherogenic. We have shown that LDL-sized particles, which are thought to be responsible for lipid accumulation within the arterial wall, cross the endothelium dominantly by a transcellular route [23,24]. It is HDL-sized particles, which are thought to be protective, and other smaller molecules that enter by the paracellular route. According to this concept, raised paracellular permeability might be beneficial. However, convective flux of water also occurs by the paracellular route in vivo and will likely affect the accumulation of macromolecules. The relations between flow multidirectionality, NO synthesis, permeability and atherosclerosis require further investigation.

## 4. Methods and Materials

### 4.1. Cell Culture

Porcine aortic endothelial cells (PAECs) were isolated using the methods of Bogle et al. [42] and cultured in Dulbecco’s modified Eagle’s medium (DMEM) supplemented with 10% foetal bovine serum (FBS), 2.5 μg/mL^−1^ amphotericin B, 100 U/mL^−1^ penicillin, 100 μg/mL^−1^ streptomycin, 50 μg/mL^−1^ gentamycin, 5 mM L-glutamine, 90 μg/mL^−1^ heparin and 5 μg/mL^−1^ endothelial cell growth factor in a humidified incubator at 37 °C under 5% CO_2_. Their purity was confirmed using DiI-acetylated LDL as previously described [20]. The medium was replaced every two days until the cells were confluent. The cells were then seeded in 12-well plates coated with biotinylated gelatine [34] and cultured until confluent again. All experiments were carried out at passage 2 because permeability increases with passage number [23].

### 4.2. Application of Shear Stress

To apply flow, the cells were cultured in multi-well plates that were swirled on the platform of an orbital shaker. This method produces a putatively proatherogenic flow in the centre of the well and a putatively antiatherogenic flow towards the edge; it permits high throughput and chronic application of shear, and is the only method in which useful numbers of cells can be exposed to multidirectional flow–see [43]. 

Wells of confluent PAEC monolayers with an average medium height of 2 mm were placed on the shaker (PSU-10i, Grant Instruments, Shepreth, UK) in an incubator and swirled for 7 days unless stated otherwise. The platform translated with an orbital diameter of 10 mm in the horizontal plane and a rotation rate of 150 rpm. Flow characteristics were obtained using computational simulations that included surface tension and wetting [33]. Static controls for each experiment were carried out using cells from the same aorta.

### 4.3. Application of Tracers

Permeability was measured using the technique of Dubrovskyi et al. [34], in which labelled avidin is added to the medium above the endothelial monolayers grown on biotinylated gelatine. The avidin binds to the biotin on crossing the endothelium; its concentration is quantified using confocal microscopy and can be related to characteristics of the overlying cells and the flow to which they were exposed [23,24,25].

The medium was replaced with DMEM supplemented as above except that the concentration of FBS was reduced to 5%. After 24 h, FITC-avidin dissolved in DMEM was applied to the wells at a final concentration of 0.38 μM, following which the wells were returned to the incubator for 3 min under swirling or control conditions. The tracer solution was then removed, and the wells were rinsed three times with PBS and fixed with 4% paraformaldehyde for 10 min. Note that the rate of macromolecular transport through the endothelium in vivo is much slower than transport within the liquid (i.e., blood) phase [44]; the same is expected in the swirling well, where substantial convection occurs. There will not be a significant concentration boundary layer, and swirling should, therefore, not affect transport through indirect mixing effects.

### 4.4. Staining

PAECs were immunostained for either cleaved caspase-9 (“early-stage apoptosis”), cleaved caspase-3 (“late-stage apoptosis”), Ki-67 (“proliferation”), phospho-Ser/Thr–Pro MPM-2 (“mitosis”) or NF-κB p65 (“inflammation”). Nuclei were stained with DRAQ5 (Biostatus), and cell borders were delineated by immunostaining for vascular endothelial (VE)-cadherin. Antibodies, labels and dilutions are listed in Table 3. In all the cases, fixed PAECs were permeabilised and blocked at room temperature with 0.1% Triton X and 2% bovine serum albumin solution for 1 h. After blocking, they were incubated with the primary antibody overnight at 4 °C followed by 1 h incubation with the secondary antibody at room temperature.

### 4.5. Inhibition of Nitric Oxide Production

N_ω_-nitro-L-arginine methyl ester (L-NAME, 500 μM final concentration) was added to the medium 24 h prior to the addition of a tracer. The monolayers being sheared were returned to the orbital shaker after the addition.

### 4.6. Imaging and Image Processing

Images were obtained with a Leica SP5 inverted confocal microscope using a ×10 0.40 NA objective and subsequently processed with MATLAB 2017a code unless stated otherwise. For display purposes, some static cultures were imaged in 96-well plates rather than in 12-well plates.

#### 4.6.1. Quantification of Tracer Accumulation and Frequency of Apoptotic and Mitotic Cells

The Leica tile scanning function was used to image along the radius of the well up to a distance of 8 mm from the centre. Areas beyond this were excluded as the time-average wall shear stress—TAWSS—is approximately constant for the distance up to 8 mm [33], and the primary aim was to investigate effects of multidirectionality of shear. Variation in indices of multidirectionality is discussed further below.

For each tile, a stack of images was acquired from three images below to three images above the monolayer, and the average intensity was calculated in each vertical column. FITC-avidin accumulation occurred in discrete patches termed “hotspots”. Thresholding by both intensity and area was used to distinguish tracer fluorescence from background noise. The resulting binarised images were overlaid on the original tile and used as a mask to quantify tracer accumulation. Signal in the absence of a tracer was negligible.

Cells stained for apoptosis or proliferation markers were similarly segmented and summed to give the number within each tile. For spatial co-registration of tracer accumulation with stained cells, a circular region of interest (ROI) with a radius of 25 μm (slightly bigger than a single cell) was centred on the centroid of each segmented cell. Images containing the circular ROIs were then overlaid on the corresponding processed FITC-avidin images to measure the extent of tracer accumulation within the ROI.

#### 4.6.2. Quantification of NF-κB p65 Translocation

Radial tile scans of PAECs stained for nuclei, VE-cadherin and NF-κB p65 were used to quantify translocation. VE-cadherin staining, indicating cell borders, was quantified using the deep residual convolutional neural network proposed by Quan et al. [45] for the segmentation of neuronal structures, implemented with the Keras open-source deep learning library. The model was trained with data that consisted of pairs of input and output images, where the input was an image of the monolayer stained for VE-cadherin and the output was manually segmented cell borders. Data augmentation—namely, 90°, 180° and 270° rotations, along with their vertical reflections—was used to supplement the training set. Training was halted when the mean squared error between model predictions and ground truth reached a plateau. The trained model was validated on a held-out test set.

DRAQ5-stained nuclei were segmented from the background using intensity and area thresholding as above. The segmented cell boundaries and nuclear images were then used as masks to quantify the mean NF-κB p65 pixel intensity value within each nucleus and within the cytoplasm of the corresponding cell. Translocation was then calculated for each cell as follows:$$\mathrm{NF}-\kappa B\;p65\;\mathrm{Translocation}=\frac{\mathrm{Mean}\;\mathrm{nuclear}\;\mathrm{NF}-\kappa B\;p65\;\mathrm{intensity}\;}{\mathrm{Mean}\;\mathrm{cytoplasmic}\;\mathrm{NF}-\kappa B\;p65\;\mathrm{intensity}}$$

For the more extensive studies investigating NF-κB p65 translocation as a function of radial distance and different shear durations, a simpler method was used: nuclei were segmented by intensity and area thresholding and used as masks to quantify the mean NF-κB p65 pixel value inside the nucleus as above, but the cytoplasmic NF-κB p65 pixel value was obtained by calculating the mean pixel value of NF-κB p65 outside the nuclear mask—that is, perfect confluence was assumed. The value of NF-κB p65 translocation was obtained using the equation above. The methods are summarised in Appendix A.

#### 4.6.3. Quantification of Endothelial Tricellular Junctions

Semantic segmentation of tricellular junctions and quantification was performed as previously described [25]. Briefly, the RefineNet architecture [46] was used to train a model to identify tricellular junctions from VE-cadherin-stained images (Appendix A), and permeable tricellular junctions were identified based on colocalization of FITC-avidin tracer accumulation and tricellular junctions. The data were subsequently used to analyse the ratio of permeable tricellular junctions to the total number of tricellular junctions, the total tracer accumulation at tricellular junctions with respect to the total transport through the whole monolayer, and the mean tracer transport through each permeable tricellular junction.

### 4.7. Statistical Methods

The results were assessed with Student’s paired t-test or the Pearson product moment correlation (r); *p* < 0.05 was used as the criterion for significance; *ns* indicates non-significance, and *, **, *** and **** denote *p* < 0.05, *p* < 0.01, *p* < 0.001 and *p* < 0.0001, respectively.

## Figures and Tables

**Figure 1 ijms-23-08076-f001:**
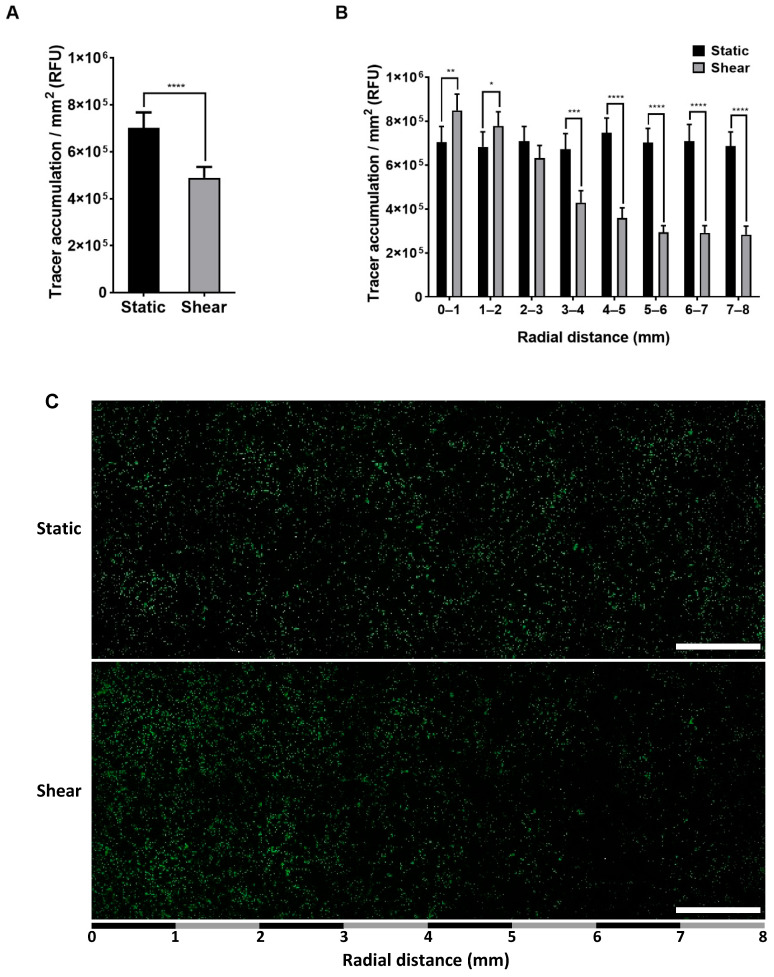
FITC-avidin accumulation, measured in relative fluorescence units (RFU), underneath static and sheared monolayers (**A**) for the entire well and (**B**) as a function of radial distance from the centre of the well. Mean + SEM, *n* = 8 isolations. (**C**) Representative tile scan of FITC-avidin accumulation (green spots) across a static well and a sheared well. Bar = 1 mm.

**Figure 2 ijms-23-08076-f002:**
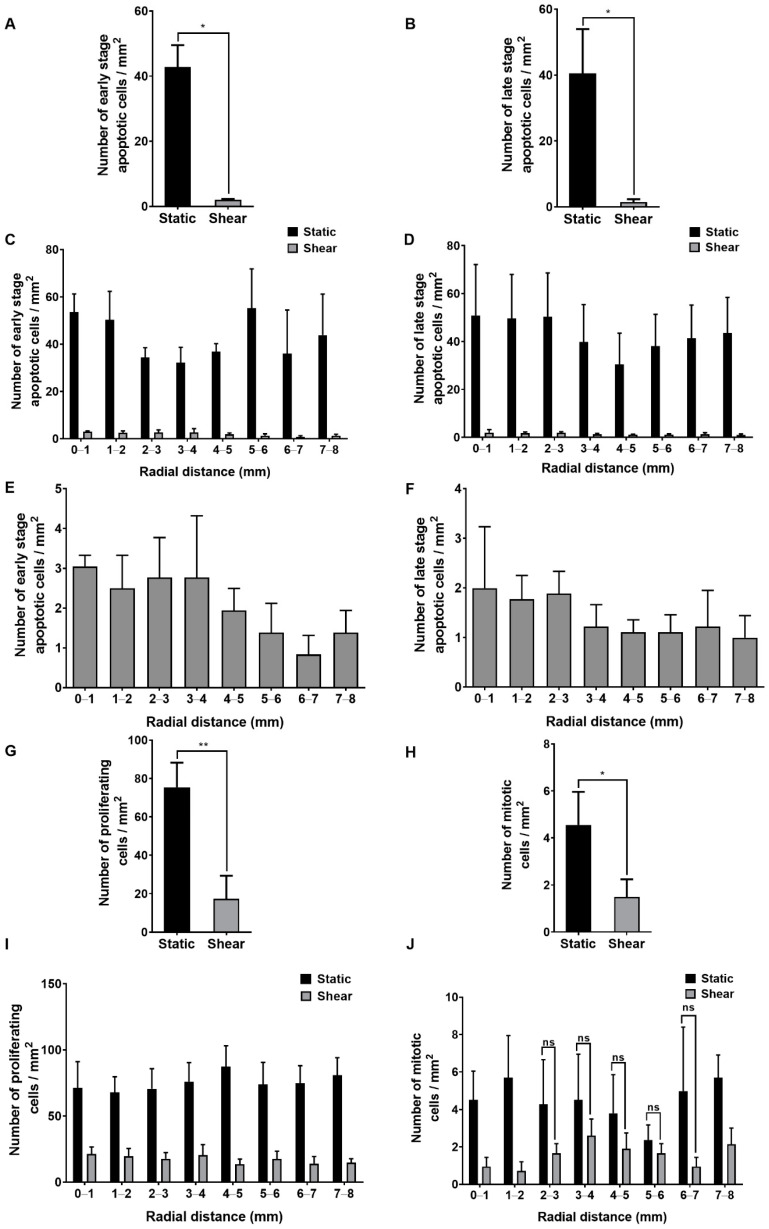
Frequency of endothelial cells in (**A**) early- and (**B**) late-stage apoptosis averaged across the entire well for static and sheared monolayers. There were approximately 750 cells/mm^2^ on the base of the well. Frequency of (**C**) early-and (**D**) late-stage apoptotic cells as a function of radial distance from the centre of the well under both conditions. Sheared results from (**C**,**D**) are presented on an expanded scale in (**E**,**F**), respectively. Frequency across the entire well of (**G**) proliferating and (**H**) mitotic cells under static conditions and shear. Frequency at different radial distances from the well centre of (**I**) proliferating and (**J**) mitotic cells under both conditions. Effects of shear were significant at all radial locations for (**C**,**D**,**I**) and, unless otherwise indicated, in (**J**). Mean ± SEM, *n* = 5 isolations.

**Figure 3 ijms-23-08076-f003:**
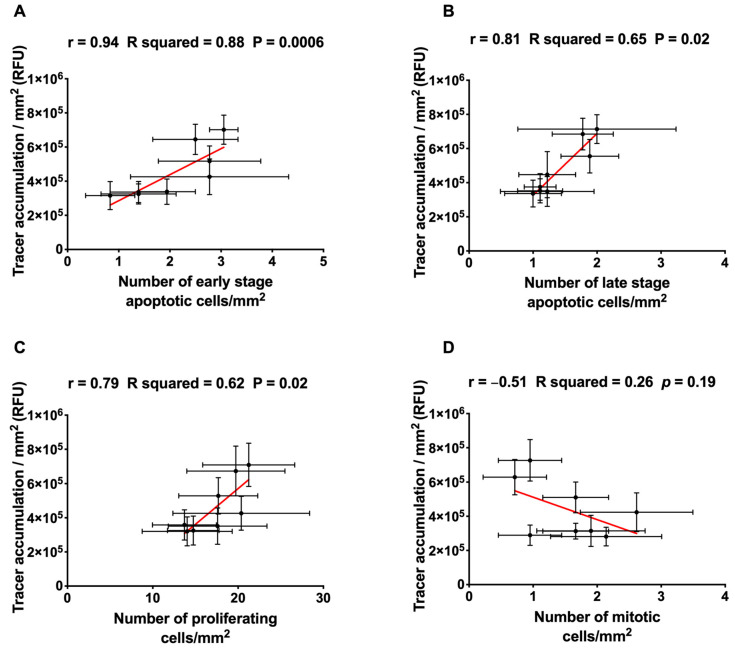
Correlation between tracer accumulation and the corresponding number of (**A**) early-stage apoptotic cells, (**B**) late-stage apoptotic cells, (**C**) proliferating cells and (**D**) mitotic cells under shear. Pearson correlation coefficient (r), R squared and *p*-value are shown above each graph. Each point represents the mean ± SEM at one radial location (*n* = 5 isolations).

**Figure 4 ijms-23-08076-f004:**
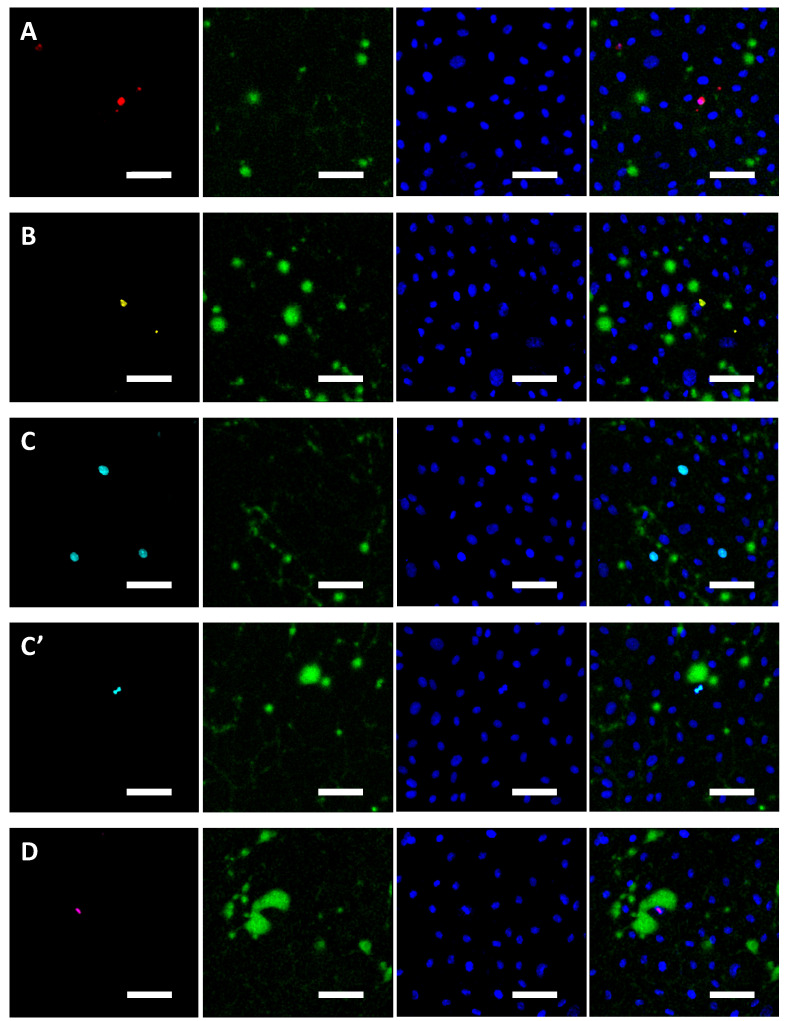
Representative confocal images of endothelial cells. The first column shows staining for (**A**) early-stage apoptosis (red), (**B**) late-stage apoptosis (yellow), (**C**) proliferation, (**C’**) a proliferating cell that is dividing (turquoise) and (**D**) mitosis (purple). The second and third columns show FITC-avidin accumulation (green) and nuclei (blue) for the same field of view. The last column shows the overlay of all three channels. Bar = 50 μm.

**Figure 5 ijms-23-08076-f005:**
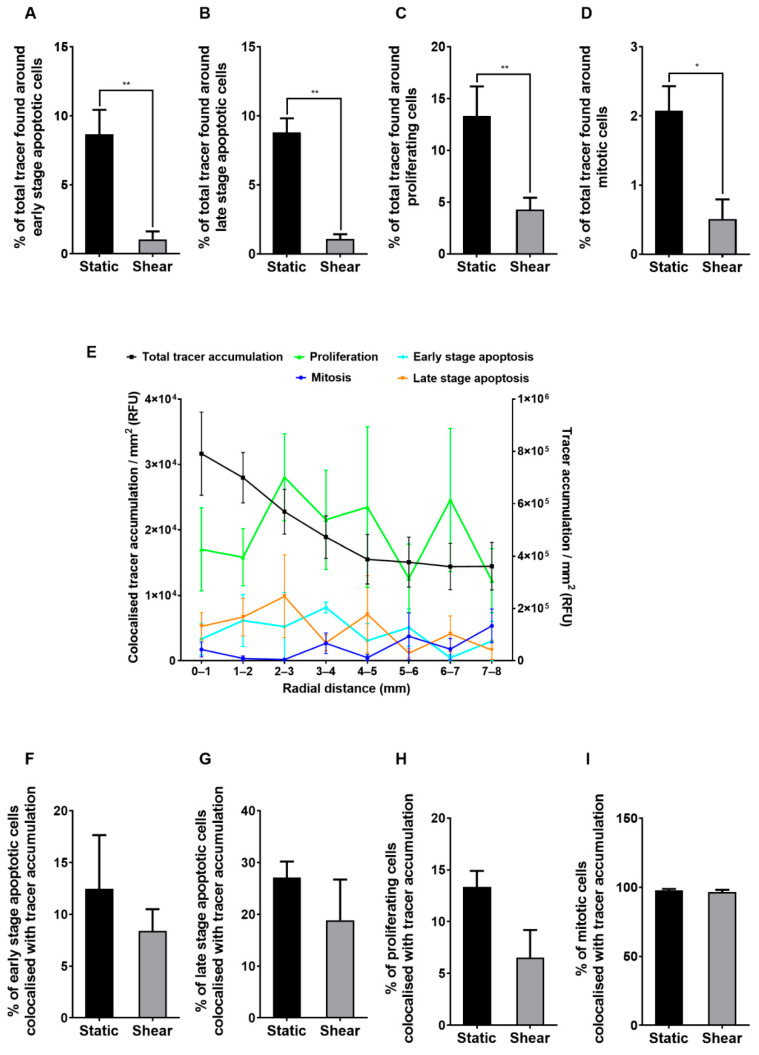
Percentage of total tracer accumulation due to cells undergoing (**A**) early-stage apoptosis, (**B**) late-stage apoptosis, (**C**) proliferation and (**D**) mitosis. (**E**) Total FITC-avidin accumulation and its accumulation near apoptotic and proliferating cells as a function of radial distance from the centre of sheared wells. Percentage of cells undergoing (**F**) early-stage apoptosis, (**G**) late-stage apoptosis, (**H**) proliferation and (**I**) mitosis which colocalised with areas of FITC-avidin accumulation under static conditions or shear. Mean ± SEM, *n* = 5 isolations.

**Figure 6 ijms-23-08076-f006:**
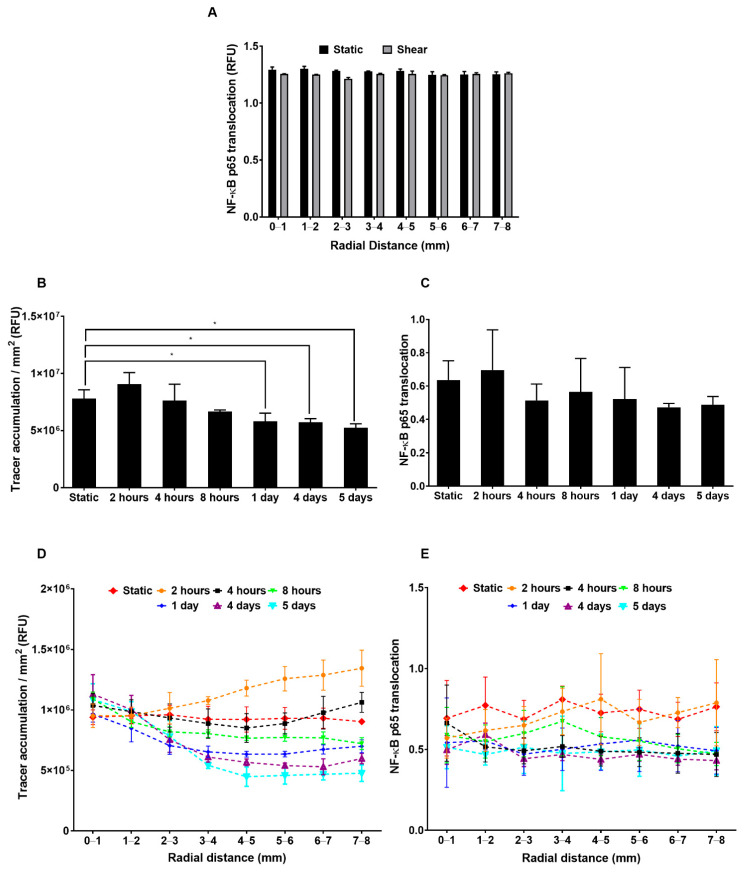
(**A**) NF-κB p65 translocation as a function of radial distance from the centre of the well at 7 days of shear or static culture. (**B**) FITC-avidin accumulation and (**C**) NF-κB p65 translocation averaged across the entire well after different durations of shear. (**D**) FITC-avidin accumulationd© (**E**) NF-κB p65 translocation as a function of radial distance from the centre of the well after different durations of shear. Mean ± SEM, *n* = 3 isolations.

**Figure 7 ijms-23-08076-f007:**
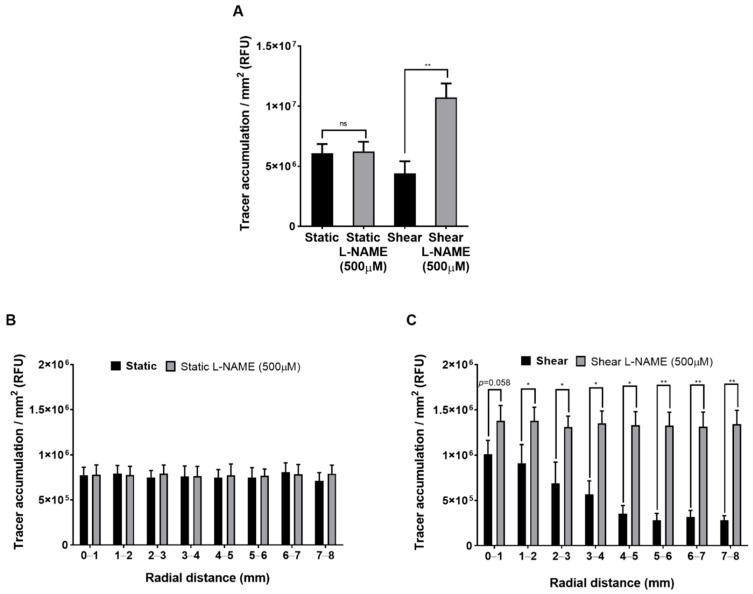
FITC-avidin accumulation averaged across the well for (**A**) static and sheared monolayers, with or without prior L-NAME treatment. FITC-avidin accumulation as a function of radial distance from the centre of the well for (**B**) static and (**C**) sheared monolayers, with or without prior L-NAME treatment. Mean + SEM, *n* = 3 isolations.

**Figure 8 ijms-23-08076-f008:**
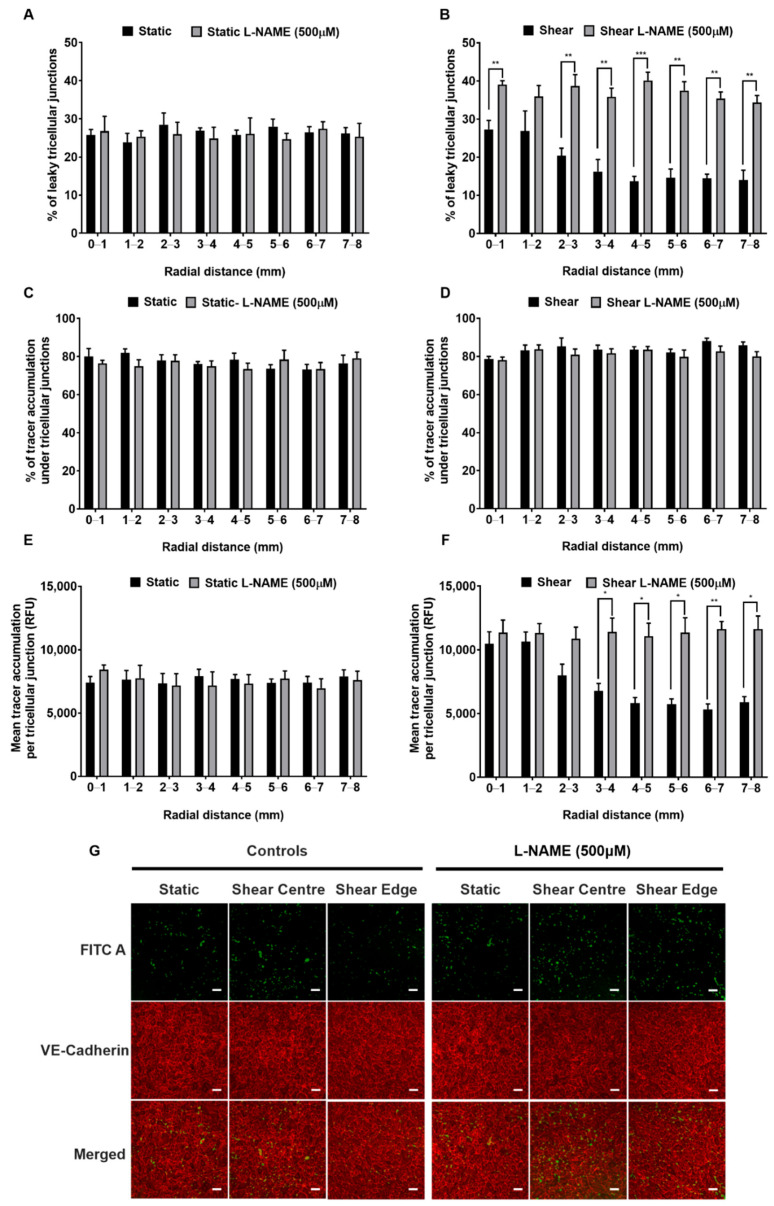
Percentage of permeable tricellular junctions of monolayers cultured under (**A**) static conditions and (**B**) shear. Percentage of the total accumulation due to tricellular junctions for the cells cultured under (**C**) static conditions and (**D**) shear. Mean tracer accumulation per permeable tricellular junction under (**E**) static conditions and (**F**) shear. All panels show data as a function of distance from the centre of the well, obtained with and without L-NAME treatment. (**G**) Confocal images of tracer accumulation (green spots) under PAEC monolayers cultured under static and sheared conditions, with and without L-NAME treatment. Red lines of anti-VE-cadherin immunostaining show cell boundaries. Bar = 100 μm. Mean ± SEM, *n* = 3 isolations.

**Table 1 ijms-23-08076-t001:** Pearson correlation coefficient (r), R squared and *p* for the total FITC-avidin uptake versus each of the colocalised datasets shown in Figure 5E.

	Early-Stage Apoptosis	Late-Stage Apoptosis	Proliferation	Mitosis
**r**	0.33	0.48	0.01	−0.48
**R squared**	0.11	0.23	0.21	0.23
** *p* ** **-value**	0.43	0.23	0.98	0.22

**Table 2 ijms-23-08076-t002:** Pearson correlation coefficie©(r), R squared and *p* for tracer accumulation versus NF-κB p65 translocation at varying radial distances from the centre of the well after different durations of shear.

	2 h	4 h	8 h	1 Day	4 Days	5 Days
**r**	0.73	0.34	0.30	0.30	0.21	0.35
**R squared**	0.53	0.12	0.0092	0.089	0.042	0.13
** *p* **	0.039	0.41	0.47	0.47	0.62	0.39

**Table 3 ijms-23-08076-t003:** List of primary and secondary antibodies used for immunofluorescence staining.

Antigen	Primary Antibody	Secondary Antibody
Cleaved caspase-9	Rabbit monoclonal anti-cleaved caspase 9, 1:800 (Cell Signalling Technology, Danvers, MA, USA)	Goat anti-rabbit Alexa Fluor 546, 1:1200 (Invitrogen, Waltham, MA, USA)
Cleaved caspase-3	Rabbit polyclonal anti-cleaved caspase 3, 1:1000 (Cell Signalling Technology)	Goat anti-rabbit Alexa Fluor 546, 1:1500 (Invitrogen)
KI67	Rabbit polyclonal anti-Ki67, 1:1000 (Abcam, Cambridge, UK)	Goat anti-rabbit Alexa Fluor 546, 1:1500 (Invitrogen)
Phospho-Ser/Thr–Pro MPM-2	Mouse monoclonal anti-phospho-Ser/Thr–Pro conjugated to Cy5, 1:800 (Merk, Darmstadt, Germany)	N/A
NF-κB p65	Rabbit polyclonal anti-NF-κB p65, 1:200 (Santa Cruz Biotechnology, Dallas, TX, USA)	Goat anti-rabbit Alexa Fluor 546, 1:300 (Invitrogen)
VE-cadherin	Goat polyclonal anti-VE-cadherin, 1:200 (Santa Cruz Biotechnology)	Donkey anti-goat Alexa Fluor 568, 1:300 (Invitrogen)

## Data Availability

Data are available from P.D.W. and M.G. on reasonable request.

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
