# Peer review of "NO Synthesis but Not Apoptosis, Mitosis or Inflammation Can Explain Correlations between Flow Directionality and Paracellular Permeability of Cultured Endothelium"

_ijms, 2022, doi:10.3390/ijms23158076_

Round 1

Reviewer 1 Report

While the authors attempt to address initial concerns, the provided fluorescent images are difficult to appreciate. Considering the dependence on fluorescent-based methods, data figures would highly benefit from high-quality representative images alongside the generated graphs (i.e. FITC-avidin, junctional marker, etc). Additional controls and labeling methods should be performed to ensure proper interpretation of the data.

Author Response

We thank the Reviewers for their valuable suggestions which have helped us to further improve our manuscript, as follows:

Reviewer 1

While the authors attempt to address initial concerns, the provided fluorescent images are difficult to appreciate. Considering the dependence on fluorescent-based methods, data figures would highly benefit from high-quality representative images alongside the generated graphs (i.e. FITC-avidin, junctional marker, etc). Additional controls and labeling methods should be performed to ensure proper interpretation of the data.

In the previous revision, we added Figure 1c that gave FITC-avidin accumulation next to the generated graphs in Figures 1a and 1b.

Supplementary Figure 1 was added to give a high-resolution image of the junctional marker VE-cadherin staining.

Supplementary Figure 2 was added to give representative images of NF-κB p65 staining.

Supplementary Figure 3 was added to give high-resolution examples of the identification of tri-cellular junctions from VE-cadherin staining.

Figure 4 already gave representative images of FITC-avidin uptake associated with apoptosis and mitosis.

Figure 7 already gave FITC-avidin uptake in association with the junctional marker VE-cadherin

We explained in the previous revision why it is technically not possible to provide images showing these properties across the whole well (i.e. to accompany the generated graphs plotting profiles across the well): the events are too sparse and too small to be visible. This is demonstrated in the following tiles scans, which show merged staining for VE-cadherin (red), NF-κB p65 (green) and nuclei (DRAQ5) across the well – nothing useful is visible. (That is why we used machine vision.)

The only thing we did not show last time was high-resolution NF-κB p65 staining, so we have now added one of the tiles from the above tilescan to Supplementary Figure 2 as follows:

Supplementary Figure 2. (Top) Outline of the methods for quantifying NF-kB p65 translocation. Representative data are shown for cells under control conditions and for cells exposed to TNF-α; more nuclear NF-kB and elongation of the cells are visible in the TNF-α case. (Bottom) A merged image of staining for NF-kB p65 (green), nuclei (blue) and VE-cadherin (red) at higher magnification.

Regarding validation, we have provided the positive control of TNFα- treatment for the NF-κB p65 staining (Supplementary Figure 2), and have also shown that our apoptosis and mitosis rates agree with previous data using different techniques (e.g. our own studies using H&E staining to identify mitotic figures).

Reviewer 2 Report

The article is very interesting, once atherosclerosis is a major cardiovascular risk factor, and is involved in the onset or development of other cardiovascular pathologies. Thus, this work is very important, and presents robust conclusions.

General comments:

  1. The objective of the work is not clear, in the abstract and in the last sentence of the introduction.
  2. The introduction is too complex, I think it could be simplified and summarized. The last sentence should be the objective of the work. I think it could simplify the sentences and make the English easier to read.
  3. What is the reason for using crops up to the 2nd passages. Normally the primary cultures are relatively stable until passage 5.
  4. the results are clear, however there are some very small graphs and it is very complex to analyze.
  5. The discussion is very complex and needs to contain the pathological explanation. That is, what is the connection of this work with vascular pathology, even if this connection is not direct, it makes sense to talk about it.

Author Response

We thank the Reviewers for their valuable suggestions which have helped us to further improve our manuscript, as follows:
 Reviewer 2
The article is very interesting, once atherosclerosis is a major cardiovascular risk factor, and is involved in the onset or development of other cardiovascular pathologies. Thus, this work is very important, and presents robust conclusions.
 General comments:
1. The objective of the work is not clear, in the abstract and in the last sentence of the introduction.
The following has been added near the end of the Abstract to emphasise the main objective and finding of the work:
“Hence shear and paracellular permeability appear to be linked by NO synthesis and not by apoptosis, mitosis or inflammation.”
Minor reductions have been made elsewhere in the Abstract, without changing its meaning, to keep within the word limit. The objective has again been added at the end of the Introduction - see next point.
2. The introduction is too complex, I think it could be simplified and summarized. The last sentence should be the objective of the work. I think it could simplify the sentences and make the English easier to read.
In the last paragraph of the Introduction, we have taken out the material describing the orbital shaker method – it has been moved to the “Application of shear stress” section of the Methods – and we have added more explanatory material. The paragraph now reads:
“Here we investigated mechanisms linking the extent of multidirectional flow to paracellular permeability. We initially focused on the roles of early- and late-stage apoptosis, proliferation and mitosis. Since the data failed to demonstrate an unequivocal link between any of these phenomena and elevated permeability at the cellular scale, we additionally studied the roles of pro-inflammatory changes and of decreased NO synthesis, both of which are early events in atherosclerosis, influenced by WSS and known to affect transport26-31. The data did not support a role for inflammation, but an inhibitor of NO synthesis abolished effects of multidirectional flow on permeability.”
3. What is the reason for using crops up to the 2nd passages. Normally the primary cultures are relatively stable until passage 5.
In our 2017 paper, we showed that endothelial permeability increases continuously with passage number:

We now say in the Methods section entitled “Cell culture”:
“All experiments were carried out at passage 2 because permeability increases with passage number23.”
4. the results are clear, however there are some very small graphs and it is very complex to analyze.
We will ensure at the proof stage that graphs are printed at an adequate size. (In the on-line version, which is how most readers access the paper these days, it is possible to magnify the figures.)
5. The discussion is very complex and needs to contain the pathological explanation. That is, what is the connection of this work with vascular pathology, even if this connection is not direct, it makes sense to talk about it.
First, we have reduced complexity. Sections of less direct relevance to our argument have been removed:
“For example, Warboys et al.20 showed that application of shear stress for 1 h increased permeability to albumin but application for 1 week decreased it; permeability under chronic shear was returned to baseline values by the nitric oxide synthase inhibitor L-NAME (and also by inhibition of phosphatidylinositol 3-OH kinase or soluble guanylyl cyclase, which are components of the NO pathway), whereas permeability under acute shear or static conditions was unaffected.”
“80% of tracer transport occurred through tricellular junctions so these phenomena are sufficient to explain our results. Nevertheless, it is of interest that the figure of 80% was unaffected at any radial location by inhibiting NO under static conditions or shear. The implication is that shear and L-NAME similarly affect bicellular junctions; these may be of only minor importance for the transport of our albumin-sized tracer, but they may be more important for smaller molecules.”
“For example, swirling could induce cells at some radial locations to release more of a permeability-increasing factor, or less of a permeability reducing factor, and that factor could then be well mixed within the medium, affecting all radial positions equally. We have already demonstrated the existence of such factors, albeit leading to a reduction in inflammation24, 42 and transcytosis24 rather than an increase in paracellular permeability.”
 (Minor clarifications and deletions have also been made – they are not shown.)
Second, we have added and deleted material in the final paragraph to make a clear and concise connection with vascular pathology:
“Finally, we consider the physiological and pathological relevance of our in vitro data. Effects of inhibiting NO production on transport in intact arteries are complex – like permeability, flow multidirectionality and the prevalence of atheromata44, they differ between regions upstream and downstream of branch points, and those differences change with age30,45,46. We previously showed that uptake of albumin is greater downstream than upstream of branch mouths in the in situ-perfused thoracic aorta of immature rabbits, and that the opposite pattern is seen in mature vessels30. Addition of an NO synthase inhibitor to the perfusate increased mean uptake in the immature vessels and reversed the pattern in the mature vessels30. A change in the same direction, albeit of smaller magnitude, was seen when NO synthesis was inhibited in mature rabbits in vivo45, and high-dose heparin, which is thought to interfere with shear-dependent NO release by disrupting the glycocalyx46, also had the same effect in vivo47. A priori, it is even hard to assert that raised paracellular permeability is pro-atherogenic. We have argued here that LDL-sized particles, which are thought to be responsible for lipid accumulation within the arterial wall, cross the endothelium dominantly by a transcellular route23,24. It is HDL-sized particles, which are thought to be protective, and other smaller molecules that enter by the paracellular route. According to this concept, raised paracellular permeability might be beneficial. However, convective flux of water also occurs by the paracellular route in vivo, and will likely affect the accumulation of macromolecules. The relations between flow multidirectionality, NO synthesis, permeability and atherosclerosis require further investigation.”

Round 2

Reviewer 1 Report

The authors have included additional images to support their initial graph quantifications.  While this study builds on previous reports by the same group using the orbital swirling system for endothelial shear studies, many of the conclusions are not robust and the L-NAME effects seems rather predictable. The novelty of this study is still very unclear. The discussion is also too lengthy and does not clearly capture the novelty of this study. The authors also indicate use of a Student’s t-test in cases where a 1-way ANOVA would be more appropriate. The text references a Fig 6F, yet this subfigure does not exist.

Author Response

see attachment for Reviewer 1

Reviewer 2 Report

The manuscript has been amended as suggested. Minor revisions are required.

In particular the objective is still not clear.

Author Response

See attached response to Reviewer 2
